# The sands of time run faster near the end

Juha Koivisto[1] & Douglas J. Durian[1]

Grains exiting an underwater silo exhibit an unexpected surge in discharge rate as they empty. This contrasts with the constant flow rate of dry granular hoppers and the decreasing flow rate of pure liquids. Here we find that this surge depends on hopper diameter and happens also in air. The surge can be turned off by fixing the rate of fluid flow through the granular packing. With no flow control, dye injected on top of the packing gets drawn into the grains. We conclude that the surge is caused by a self-generated pumping of fluid through the packing. The effect is modelled via a driving pressure set by the exit speed of the grains. This highlights a surprising and unrecognized role that interstitial fluid plays in setting the discharge rate, and perhaps in controlling clog formation, for granular hoppers whether in air or under water.

[1] Department of Physics and Astronomy, University of Pennsylvania, 3231 Walnut Street, Philadelphia, Pennsylvania 19104-6396, USA. Correspondence and requests for materials should be addressed to J.K. (email: juhakoivisto@outlook.com) or to D.J.D. (email: djdurian@physics.upenn.edu).

Hourglasses are filled with sand, rather than water, because the discharge rate of the grains is constant. In particular, it does not decrease as the filling height of the material in the upper chamber goes down, as it would for water. This feature, and the variation of discharge rate with grain and orifice size, is captured empirically by the Beverloo equations[1,2]; however, the fundamental explanation is still under active research[3–8]. The difficulty is that the grains exhibit both solid- and liquid-like behaviours near the orifice. Intuitively, the discharge speed of the grains is set by ephemeral arches that freefall though a distance proportional to the orifice size. Analogous behaviour was recently found for submerged granular hoppers, where the Beverloo equation was generalized by using the terminal falling speed[9]. But a surprising difference is that the rate is not constant unless the filling height is very large: it actually increases as the hopper empties, ever faster near the end. In contrast, the flow rate of pure liquids decreases due to decreasing hydrostatic pressure. This increasing flow rate, the surge, is important to understand as hopper flows are ubiquitous and are rarely in vacuum. Interstitial fluid affects many granular phenomena and is crucial in suspensions, fluidization and sedimentary transport[10,11]. Furthermore, the mechanisms controlling hopper discharge are basic to understand clogging and the formation of stable arches[12,13]. In a submerged granular hopper, the coupling between fluid and grains is not trivial. The viscosity and incompressibility of the fluid makes the system overdamped and grain kinetic energy dissipates into the fluid, by contrast with the underdamped highly collisional motion of grains in air or vacuum.

Here, to make progress, we show the results of our twofold experimental approach. First, we measure discharge rate versus filling height with a more precise and automated apparatus than in the previous work[9], both for submerged grains and for dry grains in air. Under these open conditions, the flow of interstitial fluid is set by the granular discharge and can self-adjust as the hopper empties. Second, we measure discharge in flow-controlled conditions where the interstitial water flow rate $Q_f^{in}$ is fixed by pumping at a range of values. The grain flow rate $Q_g$ exiting through the orifice is measured and the fluid flow rate $Q_f$ is deduced from the conservation of volume. In the flow-controlled case, we find that the granular discharge rate increases with fluid flow, but is constant in time with no surge. Using these flow-control results as input, we model the surge in the open case by combining the hydrodynamic resistance of the grains in the hopper with pressure-control set by the grain exit speed. Thus, we establish how the flow of grains is affected by the interstitial medium, and the extent to which it may not be neglected—even in air.

## Results

**Open experiment with surge.** Data for granular discharge rate $Q_g$ versus the remaining height $h$ of spherical grains in the hopper are plotted in Fig. 1a for a $D = 0.6$ cm diameter orifice and several different conditions: three different hopper diameters $D_h$ with an open top in air and under water, and one $D_h$ under a fixed input water flow rate. The quality of the data is far better than the pioneering observations, obtained by manually weighing the discharged grains within a time window of 10 s measured by a stop watch[9]. The improved automated method here records the weight continuously, 100 times faster. With this improvement we now see that the surge depends on $D_h$, and that $dQ_g/dh$ is larger for smaller $D_h$. We also discover a small surge for grains in air (inset), for which we are aware of no precedents. In accord with Beverloo, $Q_g$ in air appears constant, and independent of $D_h$, until just before this terminal surge. The third new feature in Fig. 1a is that under fixed fluid flow-control conditions, the granular

discharge rate is constant; in particular, the surge effect is totally eliminated. This indicates that the surge may be caused by interstitial fluid flow at a rate that increases as the hopper empties.

During the hopper discharge the grains near the centre of a three-dimensional cylindrical hopper flow faster than at the edges. Figure 1b is a schematic illustration along the centre plane of the hopper (Supplementary Figs 1 and 2). The dashed lines represent the conical surface of the hopper with $\theta_r$ as angle of repose. The dotted line near the orifice defines a cylinder with height $\beta D$ and diameter $D$, what we call the constricting region. Here, and only here, the grains dilate and pull the fluid from the restricting region above. The restricting bulk region merely resists the fluid motion per continuum Darcy law that connects the fluid flow and permeability of the grains.

The two experimental cases, open and flow controlled, are linked when the open case has infinitely large packing height $h$. The situation approaches the special case of the flow-controlled experiments where the superficial fluid velocity is set to match the superficial speed of grains leading to passive granular discharge rate $Q_{go}$ and passive fluid flow rate $Q_{fo}$ that are related by the packing fraction $\phi$ as $Q_{go}/\phi = Q_{fo}/(1 - \phi)$. The passive grain flow rate in the open experiments noted with primed variable $Q'_{go}$ approaches constant when the granular packing height is very high as $h$ goes to infinity as well as the hydrostatic resistance.

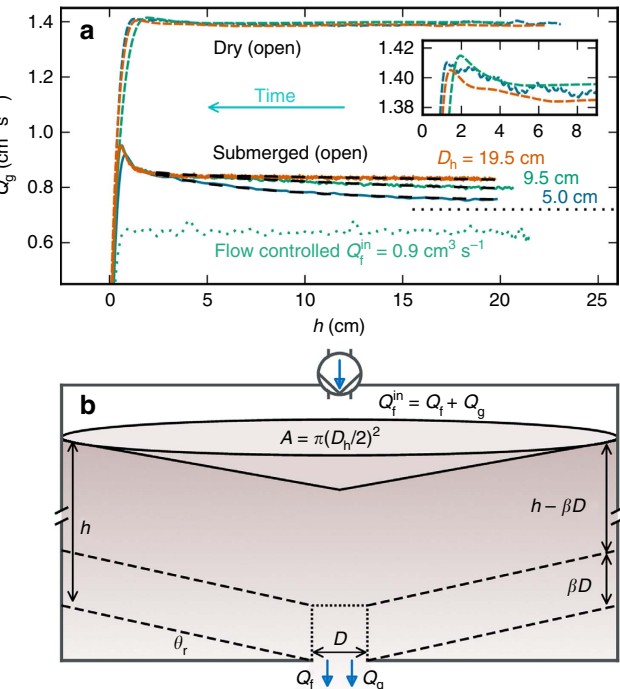

**Figure 1 | Open experiments exhibit a surge.** (**a**) Granular discharge rate versus remaining height for $d = 0.1$ cm diameter grains from $D = 0.6$ cm diameter holes, under varied conditions. The solid lines are data from submerged open experiments with black dashed curves as fits of the form $Q_g = Q'_{go} + a/(hD/A - b)$ anticipating our model shown later with $a$ and $b$ as fit parameters. These fits asymptote to the black dotted line showing the reference flow rate $Q'_{go}$ measured from an open experiment. The curves labelled with open are averages of 10 experiments each while the curve labelled flow controlled is a single set. The coloured dashed curves labelled dry correspond exactly to the open cases by geometry and analysis but without liquid. The s.d. in a single experiment of raw data is 0.1 cm$^3$ s$^{-1}$. (**b**) Schematic illustration, with defined quantities pertaining to hopper, grains and fluid.

          

To visualize the motion of interstitial fluid, we inject a layer of dye into the water just above the packing before the start of a surge experiment. Figure 2a shows discharge rate versus time, and underneath is a spacetime plot (Fig. 2b) constructed from a simultaneous digital video recording (Supplementary Movie 1). In it the grains appear brown, the dyed layer of water appears dark green and the water above the packing appears light green due to small amounts of mixed dye. The spacetime plot combines two different measurements. The top edge of the packing obtained from the camera is coincident with the data obtained from the scale shown as the white dashed curve. This result indicates that the conversion of the weight to height using a measured, constant packing fraction $\phi = 0.58 \pm 0.04$ is correct. The notion that weight and volume are proportional indicates that the amount of dilation of grains is constant and confined to the small region near the orifice, the constricting region. Later, we show that the height of this region is only few times the orifice size. For this data set, the height of the constricting region is $\beta D = 2.4$ cm.

With time, the discharge rate is seen to increase while the packing height decreases, which is seen as a slight downward tilt in the dashed line. The dyed layer is above the packing at time zero, and moves down into the packing as time progresses. And, it moves faster in tandem with the increase in discharge rate. Evidently, the act of granular discharge effectively creates a pumping effect whereby water flows down through the packing at a speed faster than the grains themselves. One might have guessed that the interstitial fluid would flow passively downward at the same speed as the grains, or slower, but in fact it moves faster. This situation contrasts with prior work on sealed containers like an hourglass[14–19], where there is an inflow of air that volumetrically matches the outflow of grains. Most recently, ref. 19 states that a common modelling approach for air inflow is an *ad hoc* modification of the Beverloo equation to include a pressure gradient opposing gravity.

**Flow-controlled experiments**. To quantify the coupling between fluid and grains, we now perform a series of flow-control measurements for how the constant grain discharge rate $Q_g$ increases with fluid input pump rate $Q_f^{in}$. The average $Q_g$ over the duration for each experiment is depicted with squares in Fig. 3a. These are well described by a linear relation $Q_g = (0.310 \pm 0.008)$ cm$^3$ s$^{-1} + (0.316 \pm 0.007) Q_f^{in}$. This fit and the error estimates use weights based on uncertainty in pump rate as well as a 1% uncertainty in $Q_g$. By volume conservation, the rate at which fluid is pumped into the top of the hopper $Q_f^{in}$ must equal the sum of grain $Q_g$ and fluid $Q_f$ flow rates exiting through the orifice: $Q_f^{in} = Q_g + Q_f$. Results for $Q_g$ may thus be recast in terms of the fluid outflow rate $Q_f$, which, as shown in Fig. 3b, is also necessarily linear in $Q_f^{in}$.

**Model and analysis**. To explain both the flow control and the surge experiments, we begin by considering the excess or deficit of fluid flow with respect to the rate $Q_{fo}$ at which the fluid flows passively with—that is, at the same speed as—the grains in the hopper. The simplest model is to assume a linear relation

$$Q_g = Q_{go} + \alpha(Q_f - Q_{fo}), \qquad (1)$$

where $Q_{go}$ is the reference grain discharge rate, when the fluid flow is passive $Q_{fo}$, and $\alpha$ is a dimensionless proportionality constant. The equation is an assumption that the fluid and grain flow rates are coupled; the excess fluid $Q_f - Q_{fo}$ and excess grain $Q_g - Q_{go}$ flow rates are proportional via $\alpha$ that can be deduced from Fig. 3.

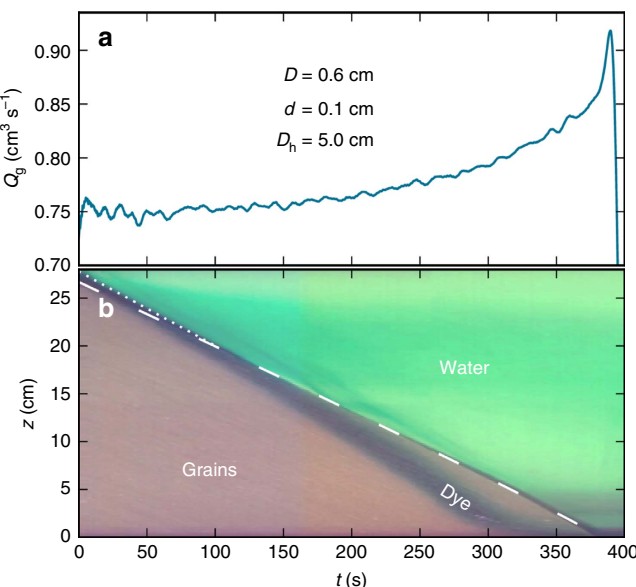

**Figure 2 | Superficial velocity of the fluid is faster than the grains'.** (**a**) Granular discharge rate versus time, and (**b**) spacetime plot from simultaneous video, for a surge experiment. The spacetime plot is constructed by taking the vertical centre line of the hopper side view from each time step $t$. Before $t = 0$, concentrated dye was injected into a layer of water above the packing. The dotted line represents the top of the dyed layer with the slope corresponding to fluid flow rate coinciding with the flow-controlled experiments. The dashed line is reproduced from the data in (**a**) by integrating the flow rate to height. The dashed curve is not a straight line but has a slight downward curvature. The dark area at the top of the granular pile is a shadow due to a dip in the centre of the packing.

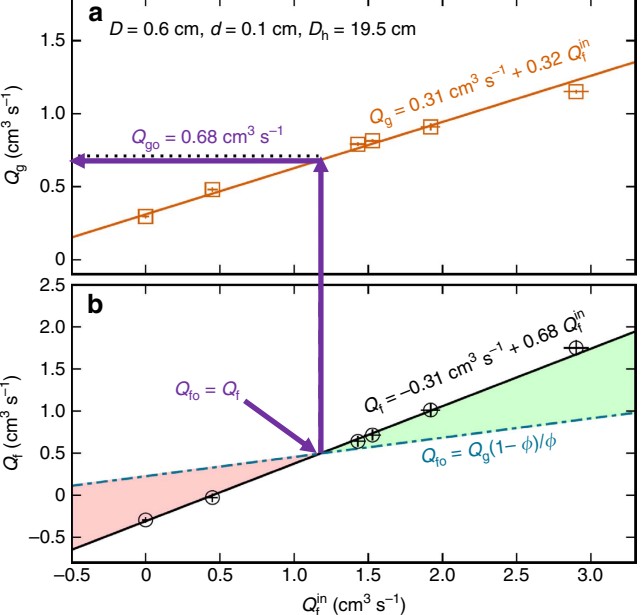

**Figure 3 | Excess and deficit fluid flow is controllable by restricting the fluid flow in.** (**a**) Granular and (**b**) fluid discharge rates versus the rate at which water is pumped in at the top $Q_f^{in}$, with standard error of the mean errorbars and line fits. The latter may be larger (green) or smaller (red) than the rate $Q_{fo}$ required for passive flow with the grains at rate $Q_g$ with equal superficial velocities. The intersect of calculated $Q_{fo}$ and measured $Q_f$ linear fits reveals the reference grain flow rate $Q_{go} = 0.68 \pm 0.05$ cm$^3$ s$^{-1}$ (indicated by arrows). A consistent value of $0.72 \pm 0.01$ cm$^3$ s$^{-1}$ (dotted line in (**a**)) is the passive flow rate $Q'_{go}$ found in the open experiments.

By volume conservation, the passive fluid flow rate is given in terms of the grain discharge rate as $Q_{fo} = Q_g(1 - \phi)/\phi$. The excess/deficit of fluid flow is indicated by the green/red shaded regions in Fig. 3b between the plotted lines for these two expressions. Their intersection graphically locates the passive reference state highlighted by the purple arrow construction. Inserting equations for volume conservation $Q_f = Q_f^{in} - Q_g$ and passive fluid flow rate $Q_{fo} = Q_g(1 - \phi)/\phi$ into equation (1), then rearranging, gives a linear dependence of $Q_g$ on $Q_f^{in}$:

$$Q_g = \left(\frac{\phi}{\alpha + \phi}\right)Q_{go} + \left(\frac{\alpha\phi}{\alpha + \phi}\right)Q_f^{in}. \quad (2)$$

Comparing with the line fit in Fig. 3a gives the two unknowns as $\alpha = 0.70 \pm 0.01$ and $Q_{go} = 0.68 \pm 0.05\,cm^3\,s^{-1}$, where the uncertainties reflects those in both the fit and in packing fraction $\phi$. The grain and orifice diameters are the same as in Fig. 1a, where the discharge rates are all higher in the open experiments than this result for flow-controlled reference flow rate $Q_{go}$, as expected. The dotted line in Fig. 3a represents the reference flow rate $Q'_{go}$ estimated from the open experiments. This same analysis of flow-control measurements is repeated for other grain and orifice sizes. The reference discharge rate obeys a modified Beverloo equation, $Q_{go} = Cv_t d^2(D/d - k)^2$, where $C = 0.4 \pm 0.1$ and $k = 2.2 \pm 1.9$, that is consistent with the preliminary study in ref. 9, and that the parameter $\alpha$ increases as $(D/d - k)^2$ (Supplementary Fig. 3). Also, the reference flow rates measured from open $Q'_{go}$ and flow-controlled $Q_{go}$ experiments are equal within error limits (Supplementary Fig. 4).

Next, we model the surge effect in an open experiment using equation (1), hydrodynamic resistance $R$ of porous medium and driving pressure $\Delta P$. The excess flow can then be expressed as $Q_f - Q_{fo} = \Delta P/R$. Since the excess fluid enters across the whole hopper area but exits through a much smaller orifice, the flow field is complex. To simplify, we approximate the medium as two porous cylinders in series: the first has area $A = \pi(D_h/2)^2$ and height $h - \beta D$, while the second has area $\pi(D/2)^2$ and height $\beta D$. Here, $\beta$ is a dimensionless parameter that sets the height of the exit region where the permeation flow constricts and the grains dilate. The total hydrodynamic resistance is then

$$R = \frac{\eta}{Kd^2}\left[\frac{h - \beta D}{A} + \gamma\frac{\beta D}{\pi(D/2)^2}\right], \quad (3)$$

where $K = (1 - \phi)^3/(180\phi^2) = 0.00122$ (Kozeny–Carman equation[20–22] for $\phi = 0.58$) and where $\gamma$ is an additional dimensionless parameter to account for the complex shape of the flow field and the increased permeability in the exit region. These ingredients for the flow resistance combine with equation (1) to predict the discharge rate versus the height $h$ of grains yet to exit as a sum of reference plus surge terms:

$$Q_g = Q_{go} + \alpha\frac{\Delta P K d^2 D/\eta}{(h - \beta D)D/A + 4\beta\gamma/\pi}, \quad (4)$$

$$= Q_{go} + \frac{a}{hD/A + b}. \quad (5)$$

Equations (4) and (5) define $a$ and $b$ as convenient fitting parameters, and give their relation to $\beta$, $\gamma$ and $\Delta P$ as the underlying unknowns. Equation (5) also highlights the form of the surge with $h$ and hopper area $A$. Namely, the surge vanishes (that is, $Q_g = Q_{go}$) in the limit $h \to \infty$, where the hydrodynamic resistance is infinite and the interstitial fluid flows passively with the grains. For smaller $h$ and larger $A$, the surge of $Q_g$ above $Q_{go}$ increases, just as seen in Fig. 1a, because the hydrodynamic resistance is smaller and the excess fluid flow is faster.

We now fit equation (5) to the three open surge experiments in Fig. 1a, by adjusting $b$ separately for each data set but adjusting

one value of $a$ and $Q'_{go}$ simultaneously for all. The quality of the fits is good, and the values of the fitting parameters make sense: First, the reference flow rate $Q'_{go} = 0.72 \pm 0.01\,cm^3\,s^{-1}$ overlaps with the result from the flow-control experiments. Second, the value of $a$ translates directly to a driving pressure of $\Delta P = a\eta/(\alpha Kd^2 D) = 5 \pm 3$ Pa. This is on the order of the Bernoulli pressure based on the single-grain terminal falling speed $v_t$, but is even closer to $\rho_f v_s^2/2$, where $v_s$ is the speed of the stream of discharged grains (Supplementary Table 1). Physically, the fluid pressure at the outlet is reduced due to some combination of grain dilation and fluid flow beneath the hopper, both of which are driven by gravity and hence might be expected to scale with $\rho_f v_t^2/2$.

Next, we repeat the surge experiments for the smaller beads and three hole and hopper sizes. Fitting values for the parameter $b$ are plotted in Fig. 4a versus $D^2/A$. As expected from equations (4) and (5), the results decrease linearly with $D^2/A$, and do not depend on grain size. The displayed line fit has slope $\beta = 4.0 \pm 0.2$ and intercept $b_o = 4\beta\gamma/\pi = 0.200 \pm 0.004$; these combine to give $\gamma = 0.04 \pm 0.01$. The height of the region where the permeation flow becomes constricted is thus $\beta D = \mathcal{O}(D)$, as expected. And, the value of $\gamma$ is smaller than one, also as expected, because the grain density decreases in the exit region and because the true interstitial flow also enters through the sidewalls of the imagined cylinder.

As a final check, we attempt to collapse all 12 surge data sets (Supplementary Fig. 5) according to equation (4), using only the fitting parameter $\beta = 4$ and the reference discharge rates $Q'_{go}$. In particular, we subtract $Q'_{go}$ and divide by the difference between $Q'_{go}$ and the rate at $h = \beta D$. The scaled discharge rates then go between 0 and 1 as $h$ decreases from infinity to $h = \beta D$. When plotted versus $x = (h - \beta D)D/A$, the data should all collapse to $b_o/(x + b_o)$. As demonstrated in Fig. 4b, the scaled data all collapse beautifully to this form with $b_o = 0.20$ as found in

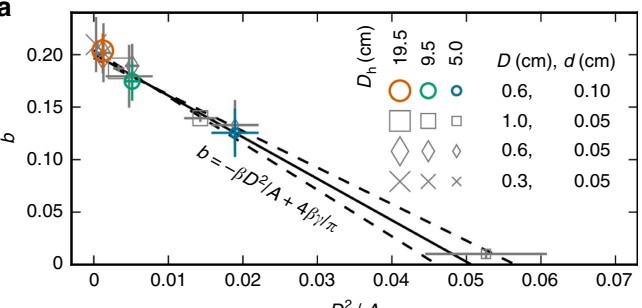

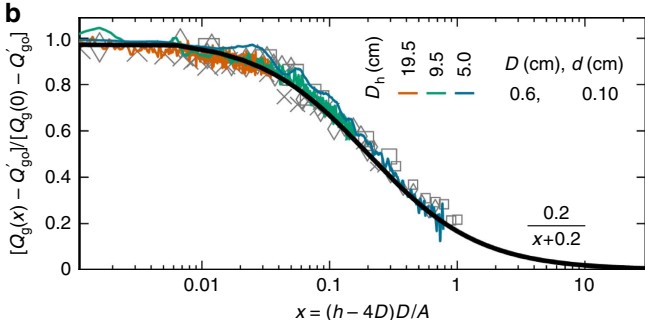

**Figure 4 | The surge data collapses with geometrical scaling.**
(**a**) Parameter $b$, from fits of equation (5) to surge data, versus $D^2/A$ with s.d. errorbars, where $A = \pi(D_h/2)^2$. (**b**) Scaled discharge data versus $x = (h - 4D)D/A$, collapsed according to equation (4). The coloured data sets are the same as in Fig. 1a.

Fig. 4a. For $h > \beta D$ the surge is thus well described and understood by our model.

## Discussion

The two-cylinder picture of the permeation flow ought to fail as $h$ decreases towards and below $\beta D$. Nonetheless, the fit to equation (5) holds well for all $h \geq \beta D$ as shown in the plots. However, for smaller $h$ there is an even more dramatic terminal surge where the discharge rate increases sharply, nearly independent of hopper diameter, as seen in Fig. 1a. This is the only effect noticeable in Fig. 1a for dry grains in air. To model the terminal surge would require a more sophisticated treatment of the shape of the packing, the permeation flow and maybe even the motion of the grains.

The model proposed here agrees with the data also at the terminal surge if the depleted resistive region is removed when calculating the hydrodynamic resistance in equation (3). Then, when $h < \beta D$ the two cylinder model then reduces to one cylinder model where $Q_g - Q_{go} \propto h^{-1}$. The experimental flow rates agree with this idea as they become independent of hopper diameter leading to an accelerating terminal surge, also in the dry case. However, more rigorous treatment of this claim would require an experiment where the height of the constricting region and the magnitude of the terminal surge is multiple decades in size. Unfortunately, this is not feasible with the current setup.

At this moment, we hypothesize that the packing fraction is lower near the orifice as the grains are already dilated due to shear stress. The final surge comes either from the rapid decrease of the hydrostatic resistance ($R \propto h$) or the mobilization of the grains. The latter is suggested by hopper simulations[8] without interstitial medium. Direct observations of this phenomenon are not possible with the current setup. Experiments in extremely high vacuum without any interstitial medium would shed light to this question.

Also, perhaps the driving pressure $\Delta P$ could no longer be treated as constant, but would increase in positive feedback with the surging discharge rate. Further studies of the fluid pressure[23] and flow[19] fields would also be helpful. In addition, it is possible that the terminal surge is an example of the faster-is-slower effect[24,25] as the fluid between the grains reduces the frictional interaction between the particles that allows clog stabilization. The flow-controlled experiments suggest that the fraction of fluid $Q_f/Q_g$ increases near the orifice as $Q_g$ increases. Similar change in packing fraction and grain motion is also seen in smaller scale when pulling liquid from granular packing with a syringe[26]. The terminal surge would then contain even more fluid between the grains and thus reducing the faster-is-slower effect.

The results presented here agree with the earlier work regarding the asymptotic behaviour of the reference flow rate and the modified Beverloo equation[9]. In addition, with the improved apparatus described here we have uncovered the full phenomenology for the surge of granular discharge as a hopper empties. We have demonstrated that this surprising effect originates in a permeation flow of interstitial fluid, which is pumped downward through the pack at a speed faster than the grains. Furthermore, we have quantitatively modelled this flow and its coupling to the granular discharge. This is significant for establishing the baseline reference state, which ought to be the target for understanding the modified Beverloo law for the discharge rate of grains under water, as a very tall hopper where the interstitial fluid flows passively downwards at the same speed as the grains. This raises a question about the usual Beverloo law for dry grains, where, both historically[1,2] and more recently[3–7], interstitial air is neglected. The baseline state of passive interstitial fluid flow may be similarly important for understanding

clogging[12], where fluid must be squeezed out from between the grains forming a stable arch over the orifice, and where pumped fluid could prevent from stable arches from forming. Creating a counter flow of fluid at the orifice prevents clogging and enables flow rates that are in the clogging regime but without clogging. Further in this regard, we now ask whether the fraction of microstates that precede a clog, as measured from the average discharge mass[13], includes grain momenta degrees of freedom that are affected by the interstitial fluid.

## Methods

**Materials.** We use monodisperse spherical glass beads of nominal diameter $d = 0.05$ cm or 0.1 cm. The grains are Potters Industries A-series technical quality glass beads. The material density is $\rho_{glass} = 2.54 \pm 0.01$ g cm$^{-3}$, found by sinking grains into water and measuring the volume of displaced water versus the increase in mass. Filtered tap water is used for the submerged cases. Based on standard textbook values for density $\rho_f$ and viscosity $\eta$, the expected terminal velocity of the smaller (larger) beads is $v_t = 7.5(15.1)$ cm s$^{-1}$, in accord with visual observation[27].

**Devices.** For all experiments we use flat-bottomed cylindrical hoppers with concentric circular orifices. The bottom plate consists of a polycarbonate disc with a 2.5 cm hole and a 5.1 cm depression to accommodate interchangeable aluminum discs 0.6 cm thick, each having an orifice of different diameter $D$. The orifice consists of a cylindrical hole that extends 0.1 cm straight down from the top of the disc and then expands out in a 45° bevel cut. The hopper sidewalls consist of interchangeable polycarbonate tubes of desired inner diameter $D_h$, and 30 cm height, glued to a flange that bolts to the bottom plate. This design makes it possible to vary the orifice diameter and hopper size reproducibly and independently. The top of the hopper is either open, or is sealed off and connected to a gear pump (Cole-Parmer 75210-50) to impose a desired volumetric flow rate $Q_f^{in}$ of water into the hopper with precision $\Delta Q_f^{in} = 0.1$ cm$^3$ s$^{-1}$.

**Analysis.** For all discharge measurements, the hopper hangs from a digital scale (Ohaus Valor 7,000, with 1 g repeatability and 10 Hz sampling rate) with continuous readout to computer, whether in air or totally submerged in a tall aquarium. The raw data is a set of mass-time pairs $m(t)$ (Supplementary Fig. 6). The mass of grains yet to be discharged is $m_g(t) = [m(t) - m_{stop}]\rho_{glass}/(\rho_{glass} - \rho_f)$, where the density factors account for buoyancy and where $m_{stop} = m(\infty)$ is the readout mass of the hopper and grains left inside when the flow stops at the end of the experiment. The height of the grains yet to be discharged is $h(t) = m_g(t)/(\rho A)$, where $\rho = 1.48 \pm 0.01$ g cm$^{-3}$ is the density of the packing, determined in an auxiliary experiment where height $h(t)$ is obtained from camera images together with mass $m_g(t)$ from the scale. The volume fraction of the packing is thus $\phi = \rho/\rho_{glass} = 0.58 \pm 0.04$, consistent with random loose packing[28]. The volumetric discharge rate of grains is $Q_g(t) = (-1/\rho_{glass})dm_g/dt$, calculated by second-degree polynomial fit over a window defined by Gaussian weighting with width $2\sigma = 6$ s (Supplementary Figs 6d, 7).

**Data availability.** Data from this study are available from the corresponding author on request.

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

## Acknowledgements

This work was supported by the Finnish Foundation's Post Doc Pool, Wihuri Foundation and Finnish Cultural Foundation (to J.K.) and by the NSF through Grant No. DMR-1305199 (to D.J.D.).

## Author contributions

J.K. performed the experiments and the data analysis. D.J.D. and J.K. designed the experiment, developed the analysis methods and the model, and wrote the article.

## Additional information

**Competing interests:** The authors declare no competing financial interests.

