## [Peer Review File · Nature Communications]

Reviewers' comments:

Reviewer #1 (Remarks to the Author):

The manuscript "the sands of time run faster near the end" aims to show that there is a distinct surge in discharge rate in submerged granular hoppers due to self-generated pumping of fluid. The authors claim that the surge, which has not been observed before, is also present in dry granular hoppers, although its characteristics are not as distinct as in the submerged case. This observation is new, and has potential applications in a variety of industrial applications in particulate handling and transport problems.

Although the topic of the manuscript is novel and distinct, I had major trouble in getting to the core of the science in this paper due to the presentation. This difficulty came from out-of-order discussion of figures (fig 1a refers to an equation on page 5, fig 1b referenced on page 5, fig 2b not referenced at all, the caption of fig 2 refers to fig 3), the lack of proper definitions of some variables (e.g. Q_f), and the lack of a proper description of the system (Q_f , Q_g , $Q_{f,in}$, $Q_{f,o}$, $Q_{g,o}$) before relations and fits are derived. Given the remit of Nature Communication (with its wide readership), I recommend strongly major revisions in the presentation of the results, as the novel observation of "increased surge towards the end" is difficult to completely unravel with the manuscript in its current state. In general, the use of the English language is good and I really appreciate the attention to detail by the authors on error margins and bounds, this is often overlooked by scientists and these authors have done this very carefully.

More general and specific comments are listed below:

- On the novelty: one author published an earlier paper in 2014 in Papers in Physics and stated: "We find that the discharge rate depends on filling height, in contrast to the well-known case of dry non-cohesive grains. It is further surprising that the rate increases up to about twenty five percent, as the hopper empties and the granular pressure head decreases." The current manuscript expands on this observation with precise experiments and further modelling, but the actual novel observation of increased discharge in wet systems has already been published elsewhere by the 2nd author.
- Abstract: the language and word choice can be sharper and more succinct for a Nature Comm. abstract, e.g. "we find ...", next sentence "we also find", or the sentence "... model this effect via a driving pressure set by the exit speed of the grains" is unnecessary complicated in structure.
- Intro: "But a surprising difference is that the rate ...": given the wide readership and varied scientific background, it is essential to distinguish explicitly between fluids and grains and state that in fluids the discharge rate is actually decreasing during hopper discharge (to highlight the surprising difference).
- Intro: a mention of the role of viscosity (in air or of other fluids) is useful here.
- Chap 1: the authors mention that the current method is "better than the pioneering observations, obtained by weighting the grains", but it is not clear to me what the current, improved method is of weighing the grains & fluid.
 - Chap 1: "small surge for grains in air": this is a novel and interesting observation, which I think the authors should expand more on. In figure 1a, the increase is not visible due to the scale, but also due to the data representation (dashed line, representing a best fit through discrete data points?). Can the authors use a separate box to magnify the discrete data and best fit, to show that this increase is beyond the noise level?
 - Chap 1: "the surge effect is totally eliminated": is this correct? The noise level seems particularly high for this measurement (percentage?), is the noise below the expected level of the surge signal, or could it be there, but hidden in the noise?
 - Chap 1: "we inject a layer a dye" \diamond "we inject a layer of dye"
 - Chap 1: Is there a relevance, or is it just a coincidence, that in the linear relation (top of page 4)

for Q_g 0.310 and 0.316 are almost equal (within error margins) and that (bottom of page 4) α is almost equal to Q_{g0} ? Is there any reason why these parameters show this remarkable similarity?

- Chap 2: "linear perturbation" \diamond "linear relationship" (it is not a perturbation).
 - Chap 2: "Next we model the surge effect" \diamond this particular sentence is long and difficult to read.
 - Chap 2: Is there a theoretical/geometrical bound on the value of β ? The authors provide a fit based on empirical data, but is there a sense why this value is true?
 - Chap 2: the driving pressure has a 60% error margin? $dP = 5 \pm 3$ Pa
 - Chap 2: is this the "faster-is-slower" effect, or is it more related to "slower-is-faster"? Perhaps the authors can expand on this in the discussion, as this is a typical topic that will appeal to a wider audience.
-
- Chap 3: what is the relevance/implications of Darcy's law? Can the authors say anything about continuum equations for this system?

Figure 1:

- Figure 1b is mute at this moment: it is not referenced and not clear as an introduction of the experimental system. Perhaps a table with definitions would help here?

Figure 2:

- The white dashed line seems to follow the peak contrast, and it seems that some dye is stuck above the grains. What is the role of dye diffusion here?
 - If Q_g increases towards the end (up to 15 – 20% according to 2a), the top of the packing should decrease more rapidly as well. However, the dashed white line does not seem to be curved. Can the authors derive quantitatively and confirm the increase in Q_f based on the slope of the dashed white curve?
- I don't understand the difference between the dotted and dashed line.
 - Can the authors derive the velocity and spreading of the dye based on the diffuse increase in the thickness of the dye layer?

Figure 3:

- This figure is difficult to comprehend and can be improved massively. What is the difference between the dotted and dashed line (after re-reading several times, I now understand it is the measured and resolved Q_{g0} , but it is vague)? The vertical dotted line needs arrows up and the crossing between Q_{f0} and Q_f needs to be highlighted by the authors, as it sets the important Q_{fin} (definitely not clear at this moment). Equation numbers instead of (rounded) fits on the lines for Q_g , Q_f and Q_{f0} would be clearer.

Figure 4:

- There is a huge gap in figure 4a between the first measurements and the last one. How sure are the authors on the fit for b over this range? Is it possible to fill the figure space with more measurement points inside the gap?

Reviewer #2 (Remarks to the Author):

This is an interesting paper describing a strange yet beautiful phenomenon with high quality experiments. The paper focuses on granular hopper flows submerged in a fluid. The authors observe several novel effects, which they model well, obtaining clear collapses with (fitted) theory. By far the most novel observation is the "pumping" effect whereby the fluid actually drains faster than the grains themselves. By considering this flow difference within a fluid resistance framework, the authors were able to model the increasing grain flux with height very clearly.

I have the following questions that should be addressed before publication is considered:

1) I'm still unsure why it is natural or obvious to expect a constant ΔP across all the experiments in Fig 1a. It would be useful for the authors to explain this. If I understand correctly, the pressure difference develops due to granular dilation during flow causing fluid to be "sucked" down. However, because the quantity of grains in the system is changing during discharge, wouldn't the amount of negative pressure depend on filling height at least to some extent? Granted, most of the dilation is near the opening, so I'd expect this wouldn't be an issue until the granular surface gets low.

2) As one goes from submerged systems to dry systems, the exponent in the Beverloo relation goes from 2 (per top of page 5) to the classical result of 5/2. Is there a simple dimensionless number, based on system inputs, that someone could use to determine if a hopper should flow at an exponent of 2 vs 5/2 (or something between)? Perhaps this is discussed in the earlier literature; if so a bit of recap would be helpful. Offhand, I might expect the numbers

$$(mg/\eta d) / (gd)^{.5} \text{ and } (mg / \rho_{\text{hof}} d^2)^{.5} / (gd)^{.5}$$

would be relevant measures of fluid-influenced particle speed (speed a particle of mass m and diameter d would achieve after moving a particle's width) relative to resistance-free free-fall speed (speed under gravity of a particle after falling a particle's width). Does one/both/neither of these numbers determine if the Beverloo relation will be in the 2 vs 5/2 regime? The reason I ask is because it would be good to know even in air what specific dimensionless measure indicates the need to account for fluid effects.

3) The paper seems to imply that dry granular flows without the effects of fluid would always flow at a constant rate, even at the end of the discharge. Does this have to be true? The paper by Dunatunga and Kamrin, utilizing a purely dry granular flow model, found a slight speed-up at the end. Also, if one re-considers the dimensional analysis giving rise to Beverloo scaling (see, e.g., Nedderman's book) the constant flow rate that comes out of the analysis relies on the assumption that the filling height is high enough not to influence Q . However, here the interesting case is for low filling height, as the hopper flow is ending, so should I really expect a constant flow rate? From a particle-scale perspective, as the hopper is almost completely discharged, the stress-supporting arch above the orifice is broken-down and eventually the flow looks like that of two heaps shedding their top layers... couldn't the diminishing of these arches over the orifice cause dry flows to speed-up near the end even without air present? This is not to suggest that one would expect a flow speed variation anywhere close to the magnitude of the one the authors observe in the wet case.

4) The authors cite reference [9] as the primary precursor study to the one here on fluid-saturated hopper flow. It would be useful for the authors to spell out a bit more clearly what this earlier study did and what the current one does new/better. Currently, the paper discusses specific differences and commonalities with this earlier paper in a couple places but it's unclear if these are the only difference/commonalities.

5) By the end of the paper it is clear the author's modeling has successfully reconciled the speed-up effect in fluid-submerged silos. However, because the final equations, (4)-(5), come from the composition of other equations and assumptions, it would be instructional to include a sentence at the end of the Model and Analysis section summarizing the physical mechanism of the speed-up as implied by the final equations.

I have the following minor points:

- Early on, it would be helpful to specify that these are 3D experiments with spherical grains.
- Φ is not defined. I assume it is packing fraction.
- What is Q'_{go} on line 114? I don't think the prime shows up before this point and I couldn't

find its definition.

- Line 52: "We inject a layer OF dye..."

I am sure the authors will be able to address my comments, at which point I can recommend publication.

Sincerely,
Ken Kamrin

Reviewer #3 (Remarks to the Author):

Dear Editor/Authors,

I have carefully reviewed the manuscript entitled "The sands of time run faster near the end". This novel work supposes a remarkable step forward in the understanding of the flow rate in submerged silos. In addition, reveals a feature of the flow rate of dry grains in silos that, despite the abundant activity in the field, has passed unnoticed over the years. Although this flow rate increase at the end of the discharge is very weak, it could be of great relevance for the sake of reaching a complete understanding of the mechanisms underlying the silo flow.

I anticipate an important impact of this work, of course within the field of granular matter, but also in other related systems concerning the flow of many particle systems through bottlenecks.

For all these reasons I think that the work should be accepted.

I have some comments that the authors might consider in order to facilitate the understanding of the manuscript.

- First, I consider that the authors should make an effort in trying to explain a bit better the first part of the Model: the one going from line 78 to 90 including the assumptions done to write equations within this part. I had to read this a few times until I get the point. Some examples: 1) ϕ is not defined; 2) "The intersection graphically locates the passive reference state". I guess that this is because at this point $Q_{fo}=Q_f$, and therefore, $Q_g=Q_{go}$, right? 3) Which are the specific equations that are inserted in eq. 1?

- The collapse shown in Fig 4b is really nice, but a couple of sentences explaining why this is so are necessary. Especially if one takes into account that the proposed model is only able to fit the behavior occurring for $h>3$ to 5 cm, i.e. the final surge is not reproduced.

- The authors speculate about the importance of their results for clogging: "where pumped fluid could break marginally stable arches". I am not really sure of this. Indeed, there are experiments where the opposite effect is observed (Phys. Rev. Lett. 92, 185506 (2004)). It is true that in this work much smaller particles are used, but the fact that this effect has been also observed in other systems, at least, questions the idea.

- Please rewrite this sentence: "Intuitively, the discharge speed of the grains is set by ephemeral arches and free-fall though a distance equal to the orifice size". Apart from being difficult to understand, equal "should" be replaced by "proportional".

- "with a more precise and automated apparatus" should be "with a more precise and automated apparatus than in the previous work".

Sincerely yours,

The referee.

Reviewer #1 (Remarks to the Author):

The manuscript "the sands of time run faster near the end" aims to show that there is a distinct surge in discharge rate in submerged granular hoppers due to self-generated pumping of fluid. The authors claim that the surge, which has not been observed before, is also present in dry granular hoppers, although its characteristics are not as distinct as in the submerged case. This observation is new, and has potential applications in a variety of industrial applications in particulate handling and transport problems.

Although the topic of the manuscript is novel and distinct, I had major trouble in getting to the core of the science in this paper due to the presentation.

RESPONSE: We thank the referee for pointing out a flaw that unintentionally sneaked into the manuscript while preparing and updating our story. The order of elements is now fixed.

This difficulty came from out-of-order discussion of figures (fig 1a refers to an equation on page 5,

RESPONSE: Equation in fig 1a caption is now written as text, not referred.

fig 1b referenced on page 5,

RESPONSE: Reference to figure 1b is removed and the fig 1b is discussed more at the beginning of the manuscript.

fig 2b not referenced at all,

RESPONSE: Reference to figure 2b added to line 77 where it belongs.

the caption of fig 2 refers to fig 3),

RESPONSE: Reference to figure 3 removed.

the lack of proper definitions of some variables (e.g. Q_f), and the lack of a proper description of the system (Q_f , Q_g , $Q_{f,in}$, Q_{fo} , Q_{go}) before relations and fits are derived.

RESPONSE: Variables (Q_f , Q_g , $Q_{f,in}$, Q_{fo} , Q_{go} , ϕ , h) are now introduced before the results between lines 35-74.

Given the remit of Nature Communication (with its wide readership), I recommend strongly major revisions in the presentation of the results, as the novel observation of "increased surge towards the end" is difficult to completely unravel with the manuscript in its current state.

RESPONSE: We have added a paragraph (lines 36-47) introducing the system better to the beginning and reworked the figure captions and equations to tackle this issue.

The only figure that we now frequently refer back to (out of order) is Figure 1a. This is the most important figure representing the data as raw as it gets in the main paper. While the data are presented in order, it also contains the fits and claims that become apparent later after we have introduced our model. We believe that showing up front how the final results relate to "raw" data makes the story stronger and helps propel the narrative.

In general, the use of the English language is good and I really appreciate the attention to detail by the authors on error margins and bounds, this is often overlooked by scientists and these authors have done this very carefully.

RESPONSE: We thank the referee for these kind words. Indeed, a great effort was placed in the analysis and their reliability.

More general and specific comments are listed below:

- On the novelty: one author published an earlier paper in 2014 in Papers in Physics and stated: "We find that the discharge rate depends on filling height, in contrast to the well-known case of dry non-cohesive grains. It is further surprising that the rate increases up to about twenty five percent, as the hopper empties and the granular

pressure head decreases.” The current manuscript expands on this observation with precise experiments and further modelling, but the actual novel observation of increased discharge in wet systems has already been published elsewhere by the 2nd author.

RESPONSE: Agreed, however there are several mitigating factors. The Paper in Physics (PIP) manuscript presented relatively crude data focused on the large-filling constant discharge rate (ie on the modified Beverloo equation), and merely pointed out the surge so that it could be avoided. It did not explore the full phenomenology of the surge: its form, how it depends on container diameter, and how it also happens for dry grain. It did not have any flow-controlled data, where the sample is sealed and fluid is pumped in. And it did not explain any of these many new observations. So there is still great novelty in the present manuscript. Another mitigating factor is that the PIP manuscript is in special edition, as a conference proceedings.

The novelty issue is also mentioned by the second referee regarding differences between Wilson et al. and this manuscript. We have elaborated the text in this regard.

- **Abstract:** the language and word choice can be sharper and more succinct for a Nature Comm. abstract, e.g. “we find ...”, next sentence “we also find”, or the sentence “... model this effect via a driving pressure set by the exit speed of the grains” is unnecessary complicated in structure.

RESPONSE: We appreciate the comment and have taken action to fix the repeats and complicated structures.

- **Intro:** “But a surprising difference is that the rate ...”: given the wide readership and varied scientific background, it is essential to distinguish explicitly between fluids and grains and state that in fluids the discharge rate is actually decreasing during hopper discharge (to highlight the surprising difference).

RESPONSE: Agreed; we added the sentence:

“In contrast, the flow rate of pure liquids decreases due to decreasing hydrostatic pressure.”

This change highlights the fluid behavior increases the readability for general audience.

- **Intro:** a mention of the role of viscosity (in air or of other fluids) is useful here.

RESPONSE: The following sentences were added:

“In a submerged granular hopper, the coupling between fluid and grains is not trivial. The viscosity and incompressibility of the fluid makes the system overdamped and grain kinetic energy dissipates into the fluid, by contrast with the underdamped highly collisional motion of grain in air or vacuum.”

- **Chap 1:** the authors mention that the current method is “better than the pioneering observations, obtained by weighting the grains ...”, but it is not clear to me what the current, improved method is of weighing the grains & fluid.

RESPONSE: The text has been clarified. "...obtained by manually weighting the discharged grains within manually determined time window of 10 seconds. The new automated method here records the change in weight of the hopper 100 times faster."

• Chap 1: "small surge for grains in air": this is a novel and interesting observation, which I think the authors should expand more on. In figure 1a, the increase is not visible due to the scale, but also due to the data representation (dashed line, representing a best fit through discrete data points?). Can the authors use a separate box to magnify the discrete data and best fit, to show that this increase is beyond the noise level?

RESPONSE: We added a magnification to figure 1a and clarified the text not to imply that there is a fit in the dry case but just in the submerged case.

• Chap 1: "the surge effect is totally eliminated": is this correct? The noise level seems particularly high for this measurement (percentage?), is the noise below the expected level of the surge signal, or could it be there, but hidden in the noise?

RESPONSE: The wording is preparing the reader for the main conclusion. The flow controlled dataset is a single measurement while the open experiments are averages of ten experiments. This explains the higher noise level. The caption is now expanded to include this.

• Chap 1: "we inject a layer a dye" \diamond "we inject a layer of dye"

RESPONSE: Fixed as suggested.

• Chap 1: Is there a relevance, or is it just a coincidence, that in the linear relation (top of page 4) for Q_g 0.310 and 0.316 are almost equal (within error margins) and that (bottom of page 4) α is almost equal to Q_{g0} ? Is there any reason why these parameter show this remarkable similarity?

RESPONSE: This is a coincidence. The parameters are related to particle and orifice diameters. For small particles per orifice ratios D/d the slope is larger and constant is smaller; vice versa for larger ratios. The particle-fluid coupling constant α increases with particle mass. The parameters contain redundancy due to volume conservation and release of potential energy to fluid. The exact relation is not trivial as seen in Figure 4 and in equation (4).

• Chap 2: "linear perturbation" \diamond "linear relationship" (it is not a perturbation).

RESPONSE: Fixed as suggested.

- Chap 2: "Next we model the surge effect" ┘ this particular sentence is long and difficult to read.

RESPONSE: True, the sentence is simplified to:

"Next we model the surge effect using Eq. (1) hydrodynamic resistance R of porous medium and driving pressure ΔP . The excess flow can then be expressed as

$$Q_f - Q_b = \Delta P / R$$

- Chap 2: Is there a theoretical/geometrical bound on the value of beta? The authors provide a fit based on empirical data, but is there a sense why this value is true?

RESPONSE: βD is the height of the constricting region depicted as solid curves in Figure 1a $\beta D = 2.4$ cm. At this point the submerged data is divided into two parts. Below 2.4 cm the data is collapsed. Above βD the data is split with the hopper diameter. This is close to the height where the surface particles are mobilized; the orifice flow is seen at the surface. Above βD the surface particles are immobile. We currently have a manuscript in preparation that shows that the ballpark is correct.

- Chap 2: the driving pressure has a 60% error margin? $dP = 5 \pm 3$ Pa

RESPONSE: Yes, unfortunately we only get the order of magnitude which is correct.

- Chap 2: is this the "faster-is-slower" effect, or is it more related to "slower-is-faster"? Perhaps the authors can expand on this in the discussion, as this is a typical topic that will appeal to a wider audience.

RESPONSE: In the "faster-is-slower" effect the particles push each other against the hopper walls and orifice edges as they rush out from the hopper and creating a blockade and slowing the flow. Here in the submerged case there is fluid between the particles that dampens the particle-particle interaction. Our measurement in Figure 3 tells us that the ratio of fluid per grain volume increases with grain flow rate. Thus, the final moments of the surge contain a lot more fluid than grains. The grains cannot interact as strongly when there is a small amount of fluid between them. We have added few sentences to paragraph just before the conclusions chapter:

In addition, it is possible that the terminal surge is an example of the "faster-is-slower" effect [AlonsoPRE12, PastorPRE15] as the fluid between the grains reduces the frictional interaction between the particles that allows clog stabilization. The flow controlled experiments suggest that the fraction Q_f/Q_g increases near the orifice as Q_g increases, also seen on Ref.~[Haw2004]. The terminal surge would then contain even more fluid between the grains and thus reducing the faster-is-slower effect.

- Chap 3: what is the relevance/implications of Darcy's law? Can the

authors say anything about continuum equations for this system?

RESPONSE: The referee has a good point. Although we are using the Darcy's law as base for our model and assume that the grains are a continuum permeable body, we don't mention this at all. We have added a sentence to the introduction on this.

Figure 1:

- Figure 1b is mute at this moment: it is not referenced and not clear as an introduction of the experimental system. Perhaps a table with definitions would help here?

RESPONSE: Agreed, we have reworked the introduction of our experiment a lot by adding two paragraphs regarding the system and the different variables as already mentioned.

Figure 2:

- The white dashed line seems to follow the peak contrast, and it seems that some dye is stuck above the grains. What is the role of dye diffusion here?

RESPONSE: We thank the referee for these observations. The figure caption is now reworked to make it more clear and contains more information.

The dark area above the grains is actually a shadow due to the dip in the top surface of the grains as well as lighting conditions and camera position. The caption is updated to include this.

The diffusion is very low compared to the fluid velocity created by the grains. This is seen when analyzing the broadening of the dark green stripe.

- If Q_g increases towards the end (up to 15 - 20% according to 2a), the top of the packing should decrease more rapidly as well. However, the dashed white line does not seem to be curved. Can the authors derive quantitatively and confirm the increase in Q_f based on the slope of the dashed white curve?

RESPONSE: Actually, the sloped dashed curve is the same data as in 2a. It is not a straight line. 20 % of curvature does not differ much from a straight line when plotted like this. We have improved the caption to clarify this.

- I don't understand the difference between the dotted and dashed line.

RESPONSE: This is related to the next question of the referee. The dotted line represents the velocity of the dye. We added clarification to the caption.

- Can the authors derive the velocity and spreading of the dye based on the diffuse increase in the thickness of the dye layer?

RESPONSE: This is related to the previous question of the referee. The dotted line is the position of the top dye layer matching the dye (=fluid) flow rate. In principle, it is possible to measure the diffusion constant from the image data. In practice the video data is not accurate enough, and the initial interface between dyed and undyed fluid is not sharp enough, for us to do so.

Figure 3:

- This figure is difficult to comprehend and can be improved massively. What is the difference between the dotted and dashed line (after re-reading several times, I now understand it is the measured and resolved Q_{g0} , but it is vague)? The vertical dotted line needs arrows up and the crossing between Q_{fo} and Q_f needs to be highlighted by the authors, as it sets the important Q_{fin} (definitely not clear at this moment). Equation numbers instead of (rounded) fits on the lines for Q_g , Q_f and Q_{f0} would be clearer.

RESPONSE: The figure is now reworked with following changes:

- Improved logic and caption
- rate Q_{go} shown with arrows.
- highlighted $Q_{fo} = Q_f$

The Q_f is important for the experimental perspective as it is our control parameter. However we feel that highlighting its importance is a bit too technical as it is just a tool to match $Q_f = Q_{fo}$ (and therefore $Q_g = Q_{go}$). Here we choose to keep the numbers in fit labels instead of the equation numbers.

Figure 4:

- There is a huge gap in figure 4a between the first measurements and the last one. How sure are the authors on the fit for b over this range? Is it possible to fill the figure space with more measurement points inside the gap?

RESPONSE: Referee raises a valid question, but we are very confident of our results as this is more due to how the data is presented. The nonlinear relation in the x-axis, D squared creates the gap. Plotting the against D instead of D^2 would decrease the gap, but the relation would not be shown as linear.

More measurements would always be possible, but similar information is already in presented in Figure 4b in different form as the data collapses. More points would just verify more of this behavior. We feel that a set with 12 different parameter combinations verifies our model. We added a reference to Supplementary Figure showing the 12 fits with the parameters in Figure 4a.

=====

Reviewer #2 (Remarks to the Author):

This is an interesting paper describing a strange yet beautiful phenomenon with high quality experiments. The paper focuses on granular hopper flows submerged in a fluid. The authors observe several novel effects, which they model well, obtaining clear collapses with (fitted) theory. By far the most novel observation is the "pumping" effect whereby the fluid actually drains faster than the grains themselves. By considering this flow difference within a fluid resistance framework, the authors were able to model the increasing grain flux with height very clearly.

I have the following questions that should be addressed before publication is considered:

1) I'm still unsure why it is natural or obvious to expect a constant ΔP across all the experiments in Fig 1a. It would be useful for the authors to explain this. If I understand correctly, the pressure difference develops due to granular dilation during flow causing fluid to be "sucked" down. However, because the quantity of grains in the system is changing during discharge, wouldn't the amount of negative pressure depend on filling height at least to some extent? Granted, most of the dilation is near the opening, so I'd expect this wouldn't be an issue until the granular surface gets low.

RESPONSE: This is a very good question but unfortunately outside of the capabilities of this setup to answer definitively. We can argue that the magnitude of the pressure does not change. However, we cannot say whether the pressure changes 20 % or so which is the change in flow rate.

The dilation of grains happens in an area that is close to the orifice. In this paper the model assumes two regions that we have named constricting region and resistive region. The constricting region has a size βD and it does not depend on height. This is seen in fig 1a as a point where data collapses without any scaling at $h = 2.4$ cm. Our model works above this limit. In the bulk, grains do not see the orifice and the packing fraction is constant. We have measured the packing fraction in an auxiliary experiment by comparing the height of the video recording and discharged mass. They are proportional with constant packing fraction (also visible in figure 2b where the dashed line of from the weight data fits on top of the figure.) We conclude that in the bulk region there is no dilation.

We have added a sentence that the packing fraction is constant and together with constant exit velocity and input velocities this leads to constant pressure at the orifice. Another possibility is that the pressure drop is generated by the large-scale flow of fluid underneath the orifice, excited by the falling grains.

2) As one goes from submerged systems to dry systems, the exponent in the Beverloo relation goes from 2 (per top of page 5) to the classical result of $5/2$. Is there a simple dimensionless number, based on system inputs, that someone could use to determine if a hopper should flow at an exponent of 2 vs $5/2$ (or something between)? Perhaps this is

discussed in the earlier literature; if so a bit of recap would be helpful. Offhand, I might expect the numbers $(mg/\eta d) / (gd)^{.5}$ and $(mg / \rho_f d^2)^{.5} / (gd)^{.5}$ would be relevant measures of fluid-influenced particle speed (speed a particle of mass m and diameter d would achieve after moving a particle's width) relative to resistance-free free-fall speed (speed under gravity of a particle after falling a particle's width). Does one/both/neither of these numbers determine if the Beverloo relation will be in the 2 vs 5/2 regime? The reason I ask is because it would be good to know even in air what specific dimensionless measure indicates the need to account for fluid effects.

RESPONSE: The referee is asking a very interesting question. Unfortunately we don't have a definite answer despite our efforts in this matter. The Beverloo equation (with exponent 5/2) and its variant (with exponent 2) apply in a steady state with high packing fractions where the fluid flow is reduced to the same superficial velocity as the grains due to infinite packing height. In the free-fall picture, both cases assume that the particles accelerate to terminal velocity and that sets the flow rate. The focus of the earlier work [Wilson et al.] is more on this topic. We hope this might be addressed by future work, perhaps in our group or by others.

3) The paper seems to imply that dry granular flows without the effects of fluid would always flow at a constant rate, even at the end of the discharge. Does this have to be true? The paper by Dunatunga and Kamrin, utilizing a purely dry granular flow model, found a slight speed-up at the end. Also, if one re-considers the dimensional analysis giving rise to Beverloo scaling (see, e.g., Nedderman's book) the constant flow rate that comes out of the analysis relies on the assumption that the filling height is high enough not to influence Q . However, here the interesting case is for low filling height, as the hopper flow is ending, so should I really expect a constant flow rate? From a particle-scale perspective, as the hopper is almost completely discharged, the stress-supporting arch above the orifice is broken-down and eventually the flow looks like that of two heaps shedding their top layers... couldn't the diminishing of these arches over the orifice cause dry flows to speed-up near the end even without air present? This is not to suggest that one would expect a flow speed variation anywhere close to the magnitude of the one the authors observe in the wet case.

RESPONSE: We thank the referee for raising a good point. Our current hypothesis is that the interstitial medium is responsible for the speedup in the terminal region.

The model here works for high packing heights. The model consists of two regions and breaks down when the upper resisting region is depleted. This just happens to be at the same point where we start to observe the final surge in air. In the submerged case the data also collapses without any scaling. Is this a coincidence?

The other hypothesis for this behavior is that the constricting region is the size of the dilating grains where there is not enough support from to form arches that shield from the weight of the upper grains. In other words, Janssen effect does not apply in the constricting region or when packing height is less than βD .

The preliminary experiments in a 0.1 atm vacuum show a surge. However this vacuum is not considered a vacuum with the current particle sizes. We have added a paragraph regarding this topic.

Perhaps the terminal surge, at the very end, is a purely granular effect as pictured by the referee in terms of heaps shedding their top layer...

4) The authors cite reference [9] as the primary precursor study to the one here on fluid-saturated hopper flow. It would be useful for the authors to spell out a bit more clearly what this earlier study did and what the current one does new/better. Currently, the paper discusses specific differences and commonalities with this earlier paper in a couple places but it's unclear if these are the only difference/commonalities.

RESPONSE: We responded to the first referee on this issue. Also, the connection between present and previous work is now expanded, both in the methods section as well as in results section. The Discussion is also reworked to highlight the differences.

5) By the end of the paper it is clear the author's modeling has successfully reconciled the speed-up effect in fluid-submerged silos. However, because the final equations, (4)-(5), come from the composition of other equations and assumptions, it would be instructional to include a sentence at the end of the Model and Analysis section summarizing the physical mechanism of the speed-up as implied by the final equations.

RESPONSE: Clarification added

I have the following minor points:

- Early on, it would be helpful to specify that these are 3D experiments with spherical grains.

Response: More details are added to section after introduction.

- Phi is not defined. I assume it is packing fraction.

RESPONSE: This is now fixed by defining ϕ as packing fraction.

- What is Q'_{go} on line 114? I don't think the prime shows up before this point and I couldn't find its definition.

RESPONSE: All variables are now described in detail.

- Line 52: "We inject a layer OF dye..."

RESPONSE: fixed as suggested.

I am sure the authors will be able to address my comments, at which point I can recommend publication.

=====

Reviewer #3 (Remarks to the Author):

Dear Editor/Authors,

I have carefully reviewed the manuscript entitled "The sands of time run faster near the end". This novel work supposes a remarkable step forward in the understanding of the flow rate in submerged silos. In addition, reveals a feature of the flow rate of dry grains in silos that, despite the abundant activity in the field, has passed unnoticed over the years. Although this flow rate increase at the end of the discharge is very weak, it could be of great relevance for the sake of reaching a complete understanding of the mechanisms underlying the silo flow.

I anticipate an important impact of this work, of course within the field of granular matter, but also in other related systems concerning the flow of many particle systems through bottlenecks.

For all these reasons I think that the work should be accepted.

I have some comments that the authors might consider in order to facilitate the understanding of the manuscript.

- First, I consider that the authors should make an effort in trying to explain a bit better the first part of the Model: the one going from line 78 to 90 including the assumptions done to write equations within this part. I had to read this a few times until I get the point. Some examples: 1) ϕ is not defined; 2) "The intersection graphically locates the passive reference state". I guess that this is because at this point $Q_{fo}=Q_f$, and therefore, $Q_g=Q_{go}$, right? 3) Which are the specific equations that are inserted in eq. 1?

RESPONSE: Thank you for the comment. The text has now been clarified clearly stating what is an assumption and how to derive the equations. The Figure 3 is clarified by adding a purple arrow construction highlighting the location of passive fluid flow.

- The collapse shown in Fig 4b is really nice, but a couple of sentences explaining why this is so are necessary. Especially if one takes into account that the proposed model is only able to fit the behavior occurring for $h > 3$ to 5 cm, i.e. the final surge is not reproduced.

RESPONSE: Correct, the model is a two-region model consisting on constricting region ($h < \beta D \sim 2$ to 5 cm) and resisting region. The model shown in figure 4b cannot work if the bulk region is depleted. However, the two region model reduces to one region model and the hydrodynamic resistance becomes proportional to packing height, the excess flow is then proportional to $1/h$. This is seen as accelerating terminal surge as well as a collapse of the data in figure 1a without any scaling with the hopper diameter, also in the dry case. Unfortunately, it is not feasible to fit $1/h$ to data with fraction of a decade in both x and y dimensions.

We have added a paragraph regarding this at the end of the results section.

- The authors speculate about the importance of their results for clogging: "where pumped fluid could break marginally stable arches". I am not really sure of this. Indeed, there are experiments where the opposite effect is observed (Phys. Rev. Lett. 92, 185506 (2004)). It is true that in this work much smaller particles are used, but the fact that this effect has been also observed in other systems, at least, questions the idea.

RESPONSE: We thank the referee for pointing out this reference that we weren't aware of. The reference is added to the manuscript.

The relation of clogging and submerged flow is not trivial. The increased liquid fraction that is induced by surge reduces clogging. By reversing the fluid flow rate at the orifice (the red region in Figure 3b), the packing fraction near the orifice is reduced. The flow rates for grains can now be below the clogging limit without a clog. We have run experiments that have not clogged with discharges of 10 kg with a typical discharge size for this flow rate in 10 g. However, the flow rate is reduced by counterflow of water and *not* by reducing the orifice. Normally clogging region is given by orifice diameter, but assuming constant flow rate it can be given as grain flow rate. Similarly, the excess flow can break arches.

We have reformulated the sentence to mean the excess flow. We have also added the reference to the introduction in the part dealing with dilation.

- Please rewrite this sentence: "Intuitively, the discharge speed of the grains is set by ephemeral arches and free-fall though a distance equal to the orifice size". Apart from being difficult to understand, equal "should" be replaced by "proportional".

RESPONSE: Agreed. The sentence is rewritten.

- "with a more precise and automated apparatus" should be "with a more precise and automated apparatus than in the previous work".

RESPONSE: Fixed as suggested.

REVIEWERS' COMMENTS:

Reviewer #1 (Remarks to the Author):

To the authors,

I am pleased to see that the authors systematically went through my suggestions point-by-point and I am satisfied with their edits. The paper is now much easier to read and I recommend the paper for publications.

A few minor issues remain:

- The quality of the data is far better than the pioneering observations, obtained by manually weighting the discharged grains within manually determined time window of 10 seconds.  "within a manually determined time window"? I am also wondering whether the second "manually" is correct. Are you referring to a "predetermined" or "selected" time window?
- when the granular packing height is very high $h \rightarrow \infty$ as well as ...  either use the infinity sign, or create a sentence (is very high as h goes to infinity)
- And we have quantitatively modelled  Grammatically it is better to start the sentence with "furthermore" or equivalent instead of "and".

Sincerely,
Nathalie Vriend

Reviewer #2 (Remarks to the Author):

The revised article and response comments have addressed my questions well. I recommend publication.

Reviewer #3 (Remarks to the Author):

Dear Editor,

I have read the resubmitted version of the manuscript "The sand of time run faster near the end" which I consider should be accepted for publication. The authors have made an important effort in order to smooth the readability, facilitating the understanding of the results. I only have some minor comments about writing:

- This increasing flow rate, the "surge" is important to understand as hopper flows -> This increasing flow rate, the "surge", is important to understand as hopper flows
- Furthermore, the mechanisms controlling hopper discharge are basic to clogging -> Furthermore, the mechanisms controlling hopper discharge are basic to understand clogging
- collisional motion of grain in air or vacuum -> collisional motion of grains in air or vacuum
- The scaled discharge rates then go between 0 and 1 as the h decreases from infinity to $h = BD$ -> The scaled discharge rates then go from 0 to 1 as h decreases from infinity $h = BD$
- Below $h < BD$ the two cylinder model then reduces to one cylinder model -> Then, when $h < BD$ the two cylinder model reduces to one cylinder model
- The experimental data agrees with this as data becomes independent of hopper diameter and produces an accelerating terminal surge, also in the dry case. -> The experimental flow rates agree with this idea as they become independent of hopper diameter leading to an accelerating terminal surge, also in the dry case.
- In addition, with an improved apparatus described here -> In addition, with the improved apparatus described here
- Caption Figure 1:

The solid lines are data from "open" experiments -> The solid lines are data from submerged
"open" experiments
analysis but without no liquid -> analysis but without liquid

REFEREE 1:

The quality of the data is far better than the pioneering observations, obtained by manually weighting the discharged grains within manually determined time window of 10 seconds.  "within a manually determined time window"? I am also wondering whether the second "manually" is correct. Are you referring to a "predetermined" or "selected" time window?

RESPONSE: Good point. What was meant that the timing was done by a stop watch. The text has been clarified accordingly.

when the granular packing height is very high $h \rightarrow \text{inf}$ as well as ...  either use the infinity sign, or create a sentence (is very high as h goes to infinity)

RESPONSE: Fixed as suggested.

And we have quantitatively modelled  Grammatically it is better to start the sentence with "furthermore" or equivalent instead of "and".

RESPONSE: Fixed as suggested.

REFEREE 3:

This increasing flow rate, the "surge" is important to understand as hopper flows -> This increasing flow rate, the "surge", is important to understand as hopper flows

RESPONSE: Fixed as suggested.

Furthermore, the mechanisms controlling hopper discharge are basic to clogging -> Furthermore, the mechanisms controlling hopper discharge are basic to understand clogging

RESPONSE: Fixed as suggested.

collisional motion of grain in air or vacuum -> collisional motion of grains in air or vacuum

RESPONSE: Fixed as suggested.

The scaled discharge rates then go between 0 and 1 as the h decreases from infinity to $h = BD$ -> The scaled discharge rates then go from 0 to 1 as h decreases from infinity $h = BD$

RESPONSE: Fixed as suggested.

Below h Then, when $h < BD$ the two cylinder model reduces to one cylinder model

RESPONSE: Fixed as suggested. (Then, when $h < \beta D$...)

The experimental data agrees with this as data becomes independent of hopper diameter and produces an accelerating terminal surge, also in the dry case. -> The experimental flow rates agree with this idea as they become independent of hopper diameter leading to an accelerating terminal surge, also in the dry case.

RESPONSE: Fixed as suggested.

In addition, with an improved apparatus described here -> In addition, with the improved apparatus described here

RESPONSE: Fixed as suggested.

Caption Figure 1:

The solid lines are data from "open" experiments -> The solid lines are data from submerged "open" experiments
analysis but without no liquid -> analysis but without liquid

RESPONSE: Fixed as suggested.